# MRD in Acute Leukemias: Lessons Learned from Acute Promyelocytic Leukemia

**DOI:** 10.3390/cancers16183208

**Published:** 2024-09-20

**Authors:** David Kegyes, Praveena S. Thiagarajan, Gabriel Ghiaur

**Affiliations:** 1MedFuture Research Center for Advanced Medicine, Iuliu Hațieganu University of Medicine and Pharmacy, 400349 Cluj-Napoca, Romania; kegyesdavid70@gmail.com; 2The Sidney Kimmel Cancer Center, The Johns Hopkins University, Baltimore, MD 21205, USA; 3Taussig Cancer Institute, Cleveland Clinic, Cleveland, OH 44106, USA; pthiaga1@kent.edu

**Keywords:** acute promyelocytic leukemia, acute myeloid leukemia, acute lymphoblastic leukemia, minimal residual disease

## Abstract

**Simple Summary:**

We provide a historical perspective on the current concepts of minimal residual disease (MRD) and how it evolves from minimal residual disease to measurable residual disease. We aim to highlight the fundamental biology behind MRD—the overlap between leukemia stem cells and MRD, and the role of the bone marrow microenvironment in the persistence of MRD. We describe how our success in measuring and subsequently eliminating residual disease in acute promyelocytic leukemia will pave the way towards progress in other acute leukemias.

**Abstract:**

**Introduction:** Advances in molecular biology, polymerase chain reaction (PCR), and next-generation sequencing (NGS) have transformed the concept of minimal residual disease (MRD) from a philosophical idea into a measurable reality. **Current Treatment Paradigms and Lessons Learned from APL:** Acute promyelocytic leukemia (APL) leads the way in this transformation, initially using PCR to detect MRD in patients in remission, and more recently, aiming to eliminate it entirely with modern treatment strategies. Along the way, we have gained valuable insights that, when applied to other forms of acute leukemia, hold the potential to significantly improve the outcomes of these challenging diseases. **Does the BM Microenvironment Play a Role in MRD?:** In this review, we explore the current use of MRD in the management of acute leukemia and delve into the biological processes that contribute to MRD persistence, including its overlap with leukemia stem cells and the role of the bone marrow microenvironment.

## 1. Introduction

The modern concept of minimal residual disease (MRD) stems from centuries of medical observations. In the East, traditional Chinese medicine texts like “*Huangdi Neijing*” place emphasis on forces within the body that manifest disease [1]. As such, disease can be “dormant” or “hidden”, only to resurface years later. In the West, building on the work of Avicenna (*The Canon of Medicine*) and Girolamo Fracastoro (*Miasmas and Contagions*), the idea that relapse is caused by “seeds of disease” that are “left behind” was the central theme of medical discourses at the beginning of the 20th century [2,3]. In the last century, with the advent of microscopy and advanced pathology, the distinction between clinical and pathological remissions became better defined for various diseases. In leukemia, physicians observed that patients continued to relapse after achieving what appeared to be a morphological complete remission (CR) [4]. This led to the hypothesis that some form of the disease persisted beyond detection. With an increasing understanding of cancer biology, the idea solidified that leukemias, even when in remission by clinical and morphological standards, might still harbor residual malignant cells capable of driving relapse. The word “minimal” in MRD emphasizes its latency and lack of detectability, encapsulating the understanding that even a minimal amount of disease, if left untreated, could result in relapse. Advances in molecular biology and the development of sensitive detection techniques, such as polymerase chain reaction (PCR) in the 1980s and 1990s, shifted the focus from a purely philosophical idea of “minimal” residual disease to a measurable entity, hence the transition to the term “measurable” residual disease [5].

## 2. Current Treatment Paradigms in Acute Leukemia and the Role of MRD

### 2.1. Traditional Approaches to Determine Treatment Response in Acute Leukemia

In acute leukemias, the primary goal of induction therapy is to reduce the total leukemic cell population to below the cytologically detectable level of approximately 10^9^ cells. Complete remission (CR) is defined as the absence of circulating blasts in the peripheral blood (PB), no signs of extramedullary disease, and less than 5% blasts in the bone marrow (BM) in patients who have hematologic recovery (absolute neutrophil count ≥ 1.0 × 10^9^/L and platelet count ≥ 100 × 10^9^/L) [6]. Traditionally, cytotoxic chemotherapy, such as “7 + 3” for AML or “4-drug induction” for acute lymphoblastic leukemia (ALL) regimens, was used to decrease the tumor burden and achieve CR [6,7]. These drugs target actively dividing cells and, thus, preferentially eliminate leukemia compared to the more quiescent normal hematopoietic stem cells. At the time of count recovery, typically 4–6 weeks from the initiation of therapy, using this strategy, 60–80% of patients with AML and 80–90% of those with ALL go on to achieve CR [6,8].

### 2.2. MRD as a Measure to Predict Treatment Response

Given the heterogeneity of the disease on one hand and the “one size fits all” of induction regimens on the other hand, intermediate assessments of response were traditionally used to predict outcomes. To this end, day 8 PB morphology with day 14 BM biopsy in AML and day 15 BM flow cytometry in ALL are indicative of the likelihood of achieving CR and may be used to intensify treatment or direct post-induction consolidation therapy [9,10,11].

The upfront incorporation of novel, targeted agents into induction regimens (i.e., venetoclax, gemtuzumab ozogamicin, and FLT3-inhibitors in AML, as well as blinatumomab and BCR::ABL1 inhibitors in ALL), however, calls into question the utility of some mid-treatment assessments as predictors of overall outcomes. For instance, the use of venetoclax in combination with “5 + 2” cytarabine and anthracycline chemotherapy leads to MRD negativity in mid-treatment assessments in almost 95% of patients, yet 30% of these patients will relapse within one year [12].

### 2.3. Assessing MRD in CR

Patients who achieve CR remain at a high risk for disease recurrence, with 60–80% of them experiencing clinical relapse. Therefore, CR cannot be used to discriminate between patients who will experience relapse from those who are cured. Traditionally, the molecular and clinical features of the disease at diagnosis or during treatment were used to identify the patients in CR who were most likely to relapse, thus leading to the intensification of consolidation regimens. The development of highly sensitive methods like multiparameter flow cytometry (MFC), PCR, and more recently, next-generation sequencing (NGS), allows clinicians to detect MRD in patients who have achieved CR. Efforts are now underway to incorporate the presence or absence of MRD into the clinical decision making regarding the management of acute leukemia in CR.

Perhaps the method with the longest track record is the use of MFC to detect a leukemia-associated immunophenotype (LAIP) in patients with ALL. The high sensitivity of this method (10^−4^ to 10^−5^), combined with decades of retrospective and prospective clinical data, make MRD detection by MFC a cornerstone of the management of these patients, with implications for risk stratification, prognosis, and treatment modifications [13]. Nevertheless, data on the use of MFC to detect MRD in ALL highlight several aspects that transcend the type of acute leukemia or the method used to detect MRD. First, the timing used to perform the test plays an important role in its predictive value. Second, the correlation between MRD positivity and relapse is not perfect, because only a fraction of MRD-positive patients go on to experience relapse [14].

When it comes to AML, unfortunately, the value of MRD is not as well-established. This may be due, at least in part, to the lack of a “standard” LAIP that can easily be identified by MFC. Molecular PCR-based methods have the sensitivity and specificity to play an important role as a tool for detecting and quantifying MRD. In ALL, the use of molecular assays to detect leukemia-specific immunoglobulin chain rearrangement is beginning to position itself as an alternative to MFC [15]. In myeloid malignancies, PCR-based methods to detect MRD have a longer track record. This is best exemplified by the use of quantitative PCR (qPCR) to detect BCR::ABL1 transcripts in patients with CML. This method has been standardized and an international scale is currently used to predict the levels of response to treatment with tyrosine kinase inhibitors and inform prognosis and treatment decisions [16]. Although in AML, multiple somatic mutations can be detected by PCR, the use of this method to identify MRD and inform treatment decisions is rather limited. The detection of NPM1 mutations and, more recently, FLT3 internal tandem duplication (FLT3-ITD) are perhaps the most advanced MRD assays in clinical development. These two mutations cover only a fraction of patients with AML, and their prognostic implications are still waiting to be fully validated. However, virtually all patients with AML carry some somatic mutations that can be detected by NGS. The use of NGS at diagnosis to guide the management of AML and inform prognosis is becoming routine, given the easy access to the method and its relatively low costs. Initially, the sensitivity of NGS was rather low (10^−2^) and, thus, not amenable to being used as an MRD method. Nevertheless, NGS pipelines that have improved their sensitivity to 10^−5^/10^−6^ and beyond are currently under development, and would make NGS a truly remarkable MRD tool (Figure 1) [17,18]. If this becomes the case, new challenges will need to be overcome before such approaches have a wide clinical applicability. For example, most somatic mutations present in AML and used to assess MRD by NGS are also found in non-AML conditions, such as clonal hematopoiesis (CH) and myelodysplastic syndromes (MDSs) (Table 1). Thus, one needs to distinguish between residual somatic mutations (such as those found in clonal hematopoiesis of indeterminate potential—CHIP) and residual leukemia cells.

### 2.4. MRD-Directed Therapies

Given the strong correlation between MRD positivity and an increased risk of relapse, achieving MRD-negative CR where no residual disease is detectable, even with current sensitive methods, has become a key objective, as it is associated with a more favorable prognosis. In fact, the importance of MRD has grown so significantly that it is used to guide therapeutic decisions in many treatment protocols and clinical trials. Achieving an MRD-negative status can lead to the de-escalation of therapy in some cases, reducing potential toxicities, while MRD positivity might indicate the need for more aggressive or alternative therapies. To this end, MRD-directed therapy with blinatumomab in patients with ALL significantly decreased the rate of relapse of these patients [19]. Interestingly, however, a recent study pointed out that providing blinatumomab consolidation, even for MRD-negative patients, increased relapse-free and overall survival (95% 3-year overall survival in the blinatumomab–chemotherapy arm versus 70% in the chemotherapy-only arm) [20]. These findings highlight the imperfect overlap between the presence of MRD and clinical relapse.

Recent data clearly highlight the need for and utility of MRD assessments in AML. Risk-adapted therapy, which takes into account the presence or absence of additional mutations in core binding factor (CBF) AML patients with translocations (8;21) and incorporates MRD testing, has been shown to significantly improve the outcomes of this sub-group [21,22]. The detection of RUNX1::RUNX1T1 in MRD testing was shown to correlate with negative outcomes in several studies [22,23]. More so, retrospective analyses of registry databases and prospective randomized clinical trials clearly show that the detection of somatic mutations via NGS at the time of transplant correlates with an increased risk of relapse [24,25]. These data suggest that perhaps more intensive conditioning regimens may mitigate the higher relapse rates in patients with MRD positivity [24,25].

Furthermore, MRD positivity during first remission can help to determine if a patient should be considered for allogeneic transplantation or guide the selection of post-transplant maintenance therapies [26]. For instance, data extrapolated from the phase 3 ADMIRAL trial (NCT02421939) comparing gilteritinib versus chemotherapy for relapsed AML patients showed the significant superiority of gilteritinib maintenance after allogeneic transplantation [27]. Subsequent randomized clinical studies, however, reported that post-transplant gilteritinib is beneficial for MRD positivity only, with its progression-free and overall survival advantages being lost in MRD-negative patients [28]. These divergent observations highlight the fact that, in the absence of randomized MRD-directed clinical trials, the value of clinical observations should be interpreted with a healthy dose of academic skepticism.

In clinical practice, MRD detection in AML could lead to either relapse (70%) or cure (25–30%), however, it is prudent to be aware that the absolute quantifiable level of disease is not the sole determinant of patient outcomes. The biology of the disease and other clinical factors modify the risk associated with MRD test results. Despite the growing evidence for the role of MRD assessment in AML treatment, there remains a significant gap in our understanding of the biology of MRD. Thus, MRD-directed randomized clinical trials are necessary to bridge this gap.

## 3. Lessons Learned from APL

The history of APL is a one-of-a-kind adventure that has transformed the most aggressive form of leukemia into the most treatable. APL accounts for 5–8% of AML patients, with distinct biological, molecular, and clinical characteristics [29]. The epidemiology of APL has been thoroughly discussed in various epidemiological studies and reviews [30,31]. Clinically, APL presents several particularities, such as disseminated intravascular coagulation and other coagulopathies. These were thoroughly discussed by Hermsen and Hambley [32]. While the presenting WBC count is part of the risk stratification for APL, the percentage of blasts is not formally part of any prognostic scoring system. Immunophenotypically, positivity for CD2, CD13, CD33, CD38, CD64, CD123, and MPO and a lack of CD34 and HLA-DR expression are highly suggestive for APL. Further details about flow cytometric diagnostics and differentials of APL were discussed by Fang et al. [33].

APL was historically considered to be a “single gene disease”, since its diagnosis, pathophysiology, and treatment were centered around a balanced reciprocal translocation between chromosomes 15 and 17, t(15;17)(q24;q21), which creates the PML-RARα fusion transcript. This product leads to the repression of both RARα and non-RARα target genes, as well as the disruption of PML nuclear bodies, which dampens p53 signaling, prevents senescence, and drives perpetual proliferation and the suppression of terminal differentiation [34]. The actual mutational profile is likely more intricate, and alterations in FLT3, WT1, NRAS, or KRAS are commonly found in these patients [35]. Initially, it was believed that additional somatic mutations have no influence on the clinical treatment of APL patients. Emerging data, however, suggest that mutations such as those in FLT3 may, in fact, correlate with a less favorable prognosis, including an increased risk of relapse, even in the modern era, and particularly in patients with a high mutational burden [36].

Over the last half-century, the management of APL has seen two transformative interventions that effectively created new “eras”. The first game-changer intervention was the introduction of retinoids (especially all-trans retinoic acid—ATRA) in the early 1980s. This coincided and was catalyzed by the use of PCR to detect PML-RARα in patients with APL. The second major shift started in the early 2000s, with the use of arsenic trioxide (ATO) in the upfront management of APL. This led to the success of current chemotherapy-free regimens (ATRA plus ATO) and unprecedented outcomes of this disease.

### 3.1. Pre-ATRA Era: Eliminating MRD, More Is Not Always Better

In the pre-ATRA era, patients with M3-AML (largely but not perfectly overlapping with APL) experienced superior outcomes compared to those with non-M3 AML, with CR rates of around 70% vs. 50% for non-M3 AML and a median overall survival (OS) of almost 10 years vs. 6 months for non-M3 AML (data from SWOG trials between 1982 and 1986) [37,38,39]. These studies established the role of high-dose anthracycline in eliminating MRD (as measured by risk of relapse). Namely, of the patients that achieved CR using 180–210 mg/m^2^ of daunorubicin, only 10% experienced relapse and another 10% died in remission. On the other hand, of the patients that achieved remission using 135 mg/m^2^ of daunorubicin, 70% experienced relapse and an additional 10% died in remission. Thus, the upfront use of high-dose daunorubicin became the standard of care for APL and catalyzed further dose intensification trials to eliminate MRD and prevent relapse.

SWOG trials between 1986 and 1991 reported that dose intensification with cytarabine improved the median overall survival of non-M3 AML to about 12 months. High-dose cytarabine, however, compared to trials from 1982–1986, resulted in worse outcomes for APL patients both in terms of CR rates (47%) and median overall survival (13 months), with 30% of these patients still relapsing [40]. Interestingly, the choice of post-remission therapy in these trials, ranging from no treatment to high-dose cytarabine or allogeneic bone marrow transplant, had no positive effect on OS [40]. In light of this evidence, one conclusion that should transcend eras is that the intensification of treatment in order to eliminate MRD should be balanced against treatment toxicities. The net benefit of such interventions can only be deduced from the results of MRD-informed, randomized, controlled clinical trials.

### 3.2. The ATRA-Era: From Minimal to Measurable Residual Disease

Catalyzed by the identification of t(15;17) (q24;q21) in virtually all patients with APL in the 1970s and the mapping of RARα to the q21 band of chromosome 17, the use of retinoids in the management of this disease marked a new approach to the treatment of acute leukemias in general [41]. Instead of using cytotoxic therapy to kill leukemia cells, the pharmacologic levels of retinoids overcome the differentiation block imposed by PML-RARα and allow malignant promyelocytes to mature into polymorphonuclear neutrophils (PMNs). These neutrophils die naturally. The mechanism of action of ATRA is multifaceted and relies on the transcriptional activation of differentiation-related genes. ATRA may also activate autophagy, leading to the early death of malignant promyelocytes [42].

Since one promyelocyte can generate 30–50 PMNs, there is an initial increase in leukocytes in response to retinoids. This phenomenon and the associated clinical signs observed during treatment with retinoids in APL were initially named retinoid syndrome and, subsequently, differentiation syndrome. Nowadays, differentiation syndrome is a common occurrence during treatment with small-molecule inhibitors that target mutations commonly found in AML, such as FLT3 inhibitors, IDH inhibitors, and menin inhibitors [43].

The initial use of single-agent retinoids, particularly ATRA, in many case reports, has shown impressive CR rates, at times as high as 100%. Huang et al. presented their experience in treating 24 patients with APL with 45–100 mg/m^2^ of ATRA. “All patients achieved complete remission without developing bone marrow hypoplasia” [44]. Unfortunately, almost all patients who achieved CR with single-agent ATRA relapsed, despite continuous therapy. However, the use of intravenous liposomal ATRA prevented relapse in 30% of patients [45]. Thus, the pharmacokinetics and tissue distribution of ATRA play important roles in eliminating MRD.

The aforementioned trials performed bone marrow biopsies on patients at the time of CR, and by using reverse transcriptase PCR (RT-PCR), showed the presence of PML-RARα fusion transcripts [46]. Similarly, in patients treated with ATRA plus chemotherapy, the detection of PML-RARα by RT-PCR was highly predictive of relapse [47]. Thus, in the early 1990s, MRD changed its meaning from minimal to measurable residual disease in APL. RT-PCR positivity at the end of induction was highly predictive for relapse and predated morphologic relapse by 1–4 months.

The success of ATRA plus chemotherapy in APL was confirmed by multicenter, randomized, controlled clinical trials carried out in Europe, the USA, and Japan [48]. Molecular methods for detecting PML-RARα (either PCR or RT-PCR) became an important readout for many clinical trials [49,50]. MRD positivity had such a high positive predictive value that the term “molecular relapse” was developed to identify patients who became MRD positive and were, thus, at an impending risk of clinical relapse [51,52].

### 3.3. Arsenic Trioxide (ATO) as a Single Agent

The use of arsenic trioxide (ATO) in APL dates back to the early 1970s with the use of Ailin-1 by a Chinese research group from Harbin University [53]. Though its mechanisms of action were not initially known, ATO was increasingly used as a salvage therapy for patients with APL who relapsed after ATRA-based therapy. When comparing the two agents, single-agent ATO saw inferior CR rates, but its 3-year rate of relapse was only 25% compared to nearly 100% with single-agent ATRA [54]. Thus, perhaps ATRA is particularly effective in controlling early mortality and differentiating the bulk of APL cells, while ATO is superior in eliminating MRD.

Over recent decades, the mechanisms of action of ATO have become clearer [29]. Arsenic has a strong affinity for the cysteine residues in PML and can directly bind to multiple domains of this gene. ATO promotes the generation of reactive oxygen species (ROS), which leads to the formation of disulfide bonds between PML-RARα fusion molecules, called nuclear matrix-associated nuclear bodies. These nuclear bodies are SUMOylated and ubiquitinated with the proteasome complex, eventually being degraded [55]. Molecular evidence of ATO targeting PML is supported by the observation that point mutations in PML (C213R, A216V, L217F, and L218P) emerge during treatment with ATO and are associated with clinical resistance in patients with APL [56].

Another described mechanism of action of ATO in APL is the promotion of both apoptosis and autophagy [57,58]. Apoptosis may be caused either through the activation of the caspase cascade or by the upregulation of p53, but other cell death mechanisms have also been linked to ATO in the literature [59].

### 3.4. ATO and Chemo-Free Regimens

Since ATRA and ATO bind different parts of the PML-RARα protein, there was early evidence for molecular synergism between these agents, which provided the rationale for combining them in clinical trials. Clinical trials using ATRA/ATO combinations revolutionized APL therapy, since almost all patients achieved CR, with the rates of molecular relapse being as low as 0% [60,61]. Since single-arm trials showed the superiority of ATRA/ATO, this served as the rationale for designing randomized clinical trials comparing ATRA/ATO with ATRA and chemotherapy. These trials showed the clear superiority of ATRA/ATO in achieving CR, as well as in preventing relapse and, thus, eliminating MRD [62].

### 3.5. A Final Lesson from APL: Assessing MRD at the End of Consolidation

A high percentage of APL patients show long-term remission and no relapse, even despite being MRD-positive after induction. Trials studying the clinical impact of ATRA/chemotherapy used MRD testing after consolidation to predict outcomes. MRD positivity after consolidation invariably characterized patients at a high risk of relapse [63]. Subsequently, in randomized clinical trials comparing ATRA/ATO with ATRA/chemotherapy, the kinetics of MRD were not significantly different between the groups [60,62].

In one study, measurable RT-PCR transcripts continued to decline during consolidation in both arms from about 3-log post-induction to close to 6-logs post-consolidation, and the rate of relapse was higher in patients treated with ATRA/chemotherapy if MRD-positive post-consolidation [62]. Thus, the most recent guidelines recommend MRD testing only at the end of consolidation for APL. However, for non-APL AML, MRD testing is recommended both during induction (after two cycles) and post-consolidation [64]. MRD is surely indispensable for the optimal management of AML patients. However, there is a significant difference between MRD positivity and relapse, both in APL and non-APL AML patients.

## 4. Does the BM Microenvironment Play a Role in MRD?

### 4.1. Leukemia Stem Cells as MRD

First postulated by Fialkow et al. in the 1960s in CML, the concept of leukemia stem cells (LSCs) enjoyed renewed interest with the identification of a CD34^+^CD38^−^ population of cells present in patients with AML that is able to recapitulate the disease in xenograft models [65,66]. Since LSCs appear to have an immunophenotype similar to normal hematopoietic stem and progenitor cells (HSPCs), there was a frenzy to identify markers that distinguish these populations in general and within individual patients. Depending on their genetic subtype, LSCs can now be identified by their differential expressions of CD34, aldehyde dehydrogenase (ALDH), CD123, CD44, CD96, CLL-1, CD47, and GPR56 [67,68,69,70,71,72,73,74,75].

Most recently, by using single-cell transcriptomics and cytometry by time of flight, we are now gaining insight not only into LSCs, but even into the transition states between normal HSPCs and LSCs [76,77]. Functionally, LSCs and HSPCs share many properties, including cell cycle quiescence and self-renewal, making them prime candidates for the persistence of MRD [78]. For patients with core binding factor AML, for instance, LSCs characterized as CD34^+^CD38^−^ALDH^int^ represent a minute fraction of the tumor at diagnosis, but are enriched in CR, with their presence predicting disease relapse [79]. Furthermore, the immunophenotype of LSCs defines the biology of leukemias and correlates with prognosis. Immature phenotypes, such as CD34 positivity in AML and ALL, are associated with worse outcomes [79,80,81].

### 4.2. BME Protects Leukemia Stem Cells and Leads to MRD

The BM is a highly complex tissue in which cell extrinsic signals provided by various microenvironments (BMEs) modulate hematopoiesis to maintain blood homeostasis. LSCs and normal HSPCs rely on similar BME signals to maintain their self-renewal, dormancy, and survival, and, thus, may directly compete to occupy various BM niches [67,82,83]. Multiple BME-dependent mechanisms have been proposed to contribute to the persistence of MRD.

#### 4.2.1. BME Maintains LSCs’ Properties

Both normal HSPCs and LSCs rely on intimate interactions with the BME to maintain their stem cell properties. For instance, direct cell–cell interactions between mesenchymal stromal cells (MSCs) and LSCs through CXCL12-CXCR4 and VCAM-1/VLA-4 promote stem cell quiescence and, thus, may contribute to the drug resistance and persistence of MRD [84,85,86]. More so, prolonged exposure to physiological levels of retinoids results in a rapid loss of HSCs and LSCs during ex vivo culture [87,88]. Mesenchymal stromal cells, via the expression of CYP26, a retinoid-inactivating enzyme, protect LSCs from differentiation and maintain their properties [87,89].

#### 4.2.2. BME Provides Pro-Survival Signals

Modern therapeutic approaches in AML rely on the use of targeted therapy to control the disease burden. While this approach has a safe side effect profile, it also creates biological opportunities to bypass this therapeutic target and contribute to drug resistance. A unique form of drug resistance takes place in the BM niche, where signals from the surrounding stroma provide pro-survival mechanisms otherwise unavailable for circulating disease [83,88]. To this end, the activation of the RAS/MAPK or JAK/STAT pathway by cytokines secreted by the BME can allow for the survival of FLT3-mutated AML blasts treated with FLT3-inhibitors when they are hosted in the BM niche [88]. Similarly, AML blasts in the BM upregulate antiapoptotic proteins, such as BCL-2, and are, thus, relatively resistant to venetoclax [90]. Furthermore, mitochondria transfer between the MSCs and AML blasts via tunneling nanotubules mitigates resistance to mitochondrial stress and provides leukemic cells with a survival advantage in the BME [91].

#### 4.2.3. BME Creates Favorable Drug Pharmacokinetics

Mesenchymal stromal cells express a large repertoire of drug-metabolizing enzymes, at levels comparable to those found in hepatocytes [92]. Of those, CYP3A4 and CYP2C19 metabolize the vast majority of traditional and novel therapies used in AML. CYP3A4 activity, in particular, has been shown to play an important role in protecting AML cells from traditional chemotherapy and targeted agents such as FLT3 inhibitors [93]. Similarly, stromal-dependent CYP26 activity protects APL and non-APL blasts from pharmacological levels of ATRA and, thus, may contribute to the persistence of MRD in patients treated with single-agent ATRA [88,89].

The relationship between the BME and the malignant cells is complex and multidirectional. Not only does the BME protect leukemia blasts, but malignant cells also reorganize their surrounding microenvironment. Blasts, for instance, transform MSCs into chemoprotective islands in ALL and AML [94,95]. More so, initial treatments can change the molecular properties of the BME. As such, treatment with ATRA in APL upregulates stromal CYP26, and chemotherapy results in higher levels of drug-metabolizing enzymes in BM MSCs [89,96].

#### 4.2.4. BME Promotes Immune Escape of LSCs

It was recognized early on that immune escape is a major mechanism responsible for the persistence of MRD. In fact, most recent bone marrow transplant trends, in which an allogeneic effect is necessary for the elimination of MRD, are exemplified by the fact that autologous transplant plays no role in the consolidation of AML. Thus, the most effective and widely used therapy to eliminate MRD in AML remains the immune therapy afforded by allogeneic transplantation. Although various mechanisms have been described in the literature, the exact ones responsible for immune escape in MRD are still not fully understood [97]. That being said, data from studies on the cellular and molecular composition of the stem cell niche point towards some potential mechanisms by which the BME may create immune sanctuaries that can protect MRD. For instance, regulatory T-cell and myeloid-derived dendritic cells are important constituents of the stem cell niche and, thus, can protect malignant cells from otherwise effective immune attacks [98]. Also, in AML, the BM niche appears to block immune cell homing and enhances the migration of LSCs by altering the CXCL12–CXCR4 axis, metabolically reprogramming immune cells, inhibiting their activation, proliferation, and cytotoxic functions [99,100]. The spatial remodeling of BM niches further hampers immune cell activity, creating a hypoxic, inflammatory, yet tumor-permissive environment [101,102].

### 4.3. Targeting the BME to Eliminate MRD

Given the close relationship between MRD, LSCs, and the BME, efforts are underway to target the BME in order to eliminate LSCs and, thus, prevent disease relapse. Initial studies were centered around mobilizing LSCs from their niche by targeting either the CXCR4–CXCL12 axis or integrins and selectins. While these strategies were able to mobilize LSCs from the protective microenvironment of the bone marrow, their combination with chemotherapy did not result in significant clinical benefits in AML and ALL [99,103]. It may be that the potential benefits of eliminating MRD were offset by the added toxicities. In addition, perhaps some of the cell-intrinsic, epigenetic properties that endow drug resistance to malignant stem cells have a level of inertia, even after LSCs are mobilized from the niche [104]. Thus, further studies are needed to find the optimal timing for LSCs’ mobilization and chemotherapeutic targeting. Perhaps combining LSC mobilization approaches with targeted therapies that have more favorable side effect profiles, such as FLT3-, IDH- or menin-inhibitors, holds the key to preventing disease relapse, as none of these treatments are curative as a single agent.

In addition to mobilizing LSCs from their BM niche in order to decrease the MRD burden, disrupting the malignant niche using antiangiogenic agents (VEGF-inhibitors or anti-VEGF antibodies) or immunomodulators (lenalidomide or pomalidomide) showed promising results in phase II clinical trials [83]. More so, recent studies showed that hypomethylating agents and epigenetic regulators such as azacytidine may reverse the immunosuppression caused by LSCs in the BM and help to restore the normal function of the BM T-cells’ microenvironment [105].

Similarly, the robust immune reconstitution of the BM microenvironment plays an important role in the persistence of MRD in ALL and, thus, has a profound impact on the long-term relapse-free survival of these patients [106]. While immunotherapies have achieved remarkable clinical success in ALL, only a few antibody therapies, including CAR T-cells, NK cells, and tumor vaccines, have demonstrated activity against AML blasts. Several reviews have summarized the current clinical knowledge on these therapies and highlighted that, at this time, the most promising targets in AML appear to be CD33 and CD123 [107,108,109,110].

Lastly, drug formulations that have improved pharmacokinetic properties, such as liposomal cytarabine plus anthracycline (CPX-351), may bypass the biochemical barrier created by stromal drug-metabolizing enzymes and, thus, eliminate the LSCs in their niche. To this end, treatment with CPX-351 resulted in a high rate of MRD-negative CR, which may explain the superior survival observed compared to the traditional “7 + 3” [111]. Similarly, liposomal formulations of ATRA have a greater efficiency in preventing relapse compared to oral ATRA [88]. In addition, synthetic retinoids that are resistant to CYP26-mediated degradation, such as tamibarotene (Sy1425) and IRX195183, were also able to bypass the stromal biochemical barrier and differentiate APL and non-APL leukemia cells in preclinical and clinical settings [89,112,113]. In APL, tamibarotene had similar CR rates to ATRA, but had significantly lower rates of relapse (50% versus 100%), highlighting the importance of adequate local retinoid pharmacokinetics in the control of MRD [114].

The contribution of the BME to MRD is summarized in Figure 2. Also, the important lessons taught by APL and discussed in the previous sections are summarized in Table 2.

## 5. Conclusions

We have made remarkable strides in the management of APL, yet we often fail to rapidly translate these advancements to other forms of leukemia. It was once thought that APL is a genetically simple disease affecting younger demographics and, therefore, insights from APL biology could not be applied to other leukemias. However, we now know that APL is genetically diverse and that neither additional genetic lesions nor extremes of age significantly impact outcomes. In fact, the most crucial determinant of outcomes is treatment with ATRA.

It was also believed that the success seen with the use of ATRA for APL could not be replicated in other AML subtypes. Yet, similar to ATRA + ATO in APL, we now see that targeting driver mutations in other AMLs differentiates malignant blasts and leads to differentiation syndrome. We also know that single-agent ATRA is not effective in eliminating MRD in APL, likely due to BME-dependent mechanisms [89,103,112,115]. While it is likely that ATO degrades PML-RARA in APL MRD, other mechanisms of action have also been described for ATO, including the induction of ROS, the promotion of autophagy, and the inhibition of Hedgehog signaling [116,117,118]. To what extent these mechanisms play a role in the success of the ATRA + ATO combination in eliminating MRD remains to be explored.

Albert Einstein is often quoted as saying, “Insanity is doing the same thing over and over and expecting different results”. In the same way, it is madness to make breakthroughs in APL treatment without applying those lessons to other leukemias, yet still expect rapid progress. The question is, how long will it take for us to apply these lessons to non-APL AML?

Building on our current knowledge, future AML research should prioritize advancing MRD detection through novel assessment tools, identifying new mutations for MRD monitoring, and developing MRD prognostic scoring systems. Thus, developing novel drug combinations that target the bone marrow microenvironment hold the potential to bypass the stroma-mediated protection of LSCs and stimulate the immune system to eliminate MRD and improve long-term outcomes in patients with AML.

## Figures and Tables

**Figure 1 cancers-16-03208-f001:**
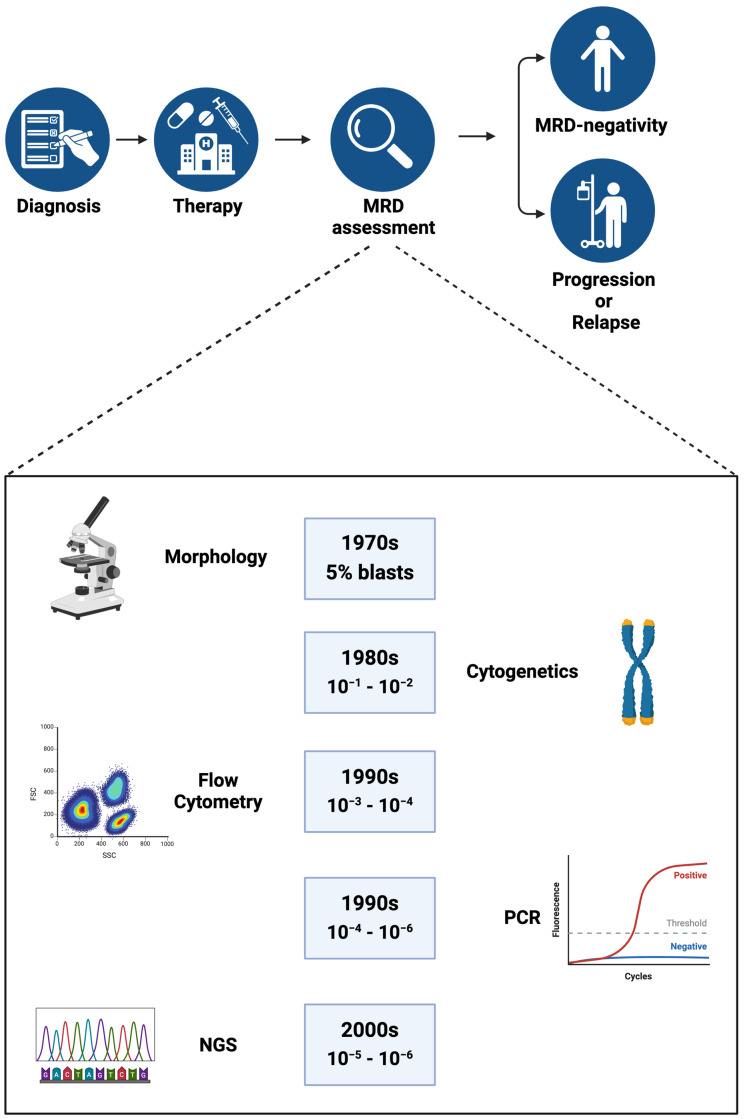
MRD assessment tools. Years of implementation and detection thresholds are illustrated for each tool. PCR—polymerase chain reaction and NGS—next-generation sequencing. Figure created in Biorender.com, accessed on 19 September 2024.

**Figure 2 cancers-16-03208-f002:**
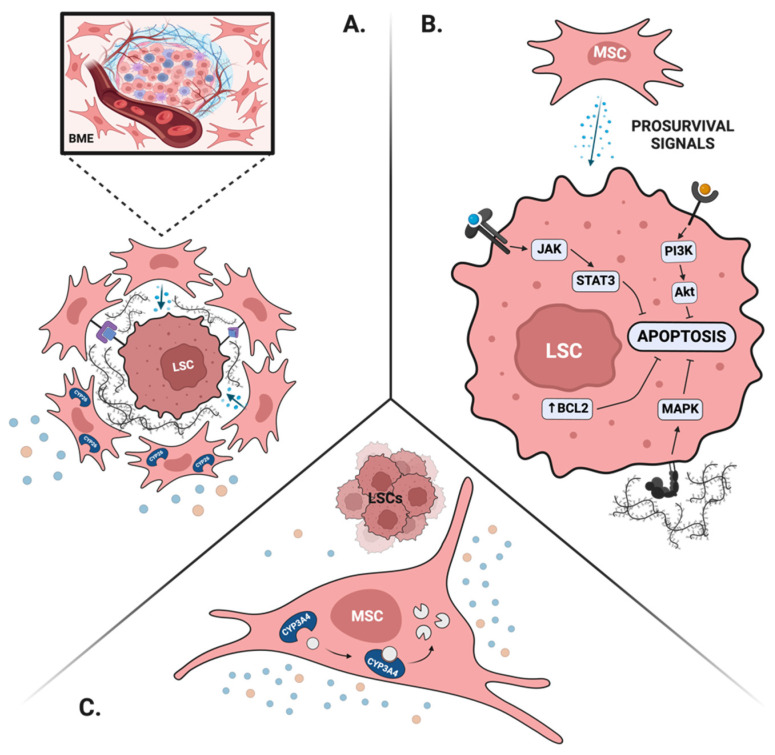
The role of the bone marrow microenvironment (BME) in persistence of MRD. (**A**) Bone marrow microenvironment creates optimal niches for maintenance of leukemia stem cells by providing extracellular matrix, soluble factors, direct cell-to-cell signaling, and protecting them from endogenous retinoids. (**B**) Soluble factors produced by MSCs activate pro-survival signaling pathways, such as the JAK/STAT, MAPK and PI3K pathways to upregulate antiapoptotic proteins, such as BCL2 in LSCs. (**C**) MSCs create biochemical barriers to chemotherapeutics via expression of drug-metabolizing enzymes. Figure created in Biorender.com, accessed on 15 August 2024.

**Table 1 cancers-16-03208-t001:** Somatic mutations associated with AML, MDS, or CHIP. All these mutations occur in AML, MDS, and CHIP, respectively; nevertheless, their incidence varies, making them more typical of one or the other.

Mutation	Disease(s) in Which More Prevalent
RAS (KRAS, NRAS)	AML, MDS, CHIP
SF3B1	AML, MDS, CHIP
SRSF2	AML, MDS, CHIP
ASXL1	AML, MDS, CHIP
DNMT3A	AML, MDS, CHIP
TP53	AML, MDS, CHIP
TET2	AML, MDS, CHIP
U2AF1	AML, MDS, CHIP
JAK2	AML, MDS, CHIP
PHF6	AML, MDS
GATA2	AML, MDS
RUNX1-RUNX1T1	AML, MDS
FLT3-ITD	AML, MDS
CEBPA	AML, MDS
EZH2	AML, MDS
IDH 1 and IDH 2	AML, MDS
ETV6	AML, MDS
PHF6	AML, MDS
ZRSR2	AML, MDS
BCOR	AML, MDS
CHEK2	CHIP
ATM	CHIP
VAF	CHIP
TERT	CHIP
SMC4	CHIP
CD164	CHIP
NPAT	CHIP
PARP1	CHIP
KDM6A	MDS
STAG2	MDS
RAD21	MDS
WT1	MDS
DEK-NUP214	AML
CBFB-MYH11	AML
NPM1	AML
MLLT3-KMT2A	AML
MECOM (EVI1)	AML

**Table 2 cancers-16-03208-t002:** Summary of lessons learned from APL and how these may be extrapolated to other leukemias.

Lessons Learned from APL	Concept	Current Trends in Leukemias
High-dose cytarabine leads to lower CR and higher relapse rates	More is not always better	Need for MRD-directed randomized clinical trials
ATRA induces differentiation syndrome	Inhibiting driver mutations leads to differentiation syndrome	FLT3-, IDH-, and menin-inhibitors induce differentiation syndrome
Single-agent ATRA remission without cure	Inhibiting driver mutations alone is not enough to eliminate MRD	FLT3-, IDH-, and menin-inhibitors cannot eliminate MRD as single agents
RT-PCR for PML-RARα	Molecular methods to detect MRD	PCR and NGS for somatic mutations (NPM1, FLT3)
CYP26 protects LSCs from ATRA Liposomal ATRA is superior to oral ATRA in preventing relapse	BME creates a biochemical barrier to protect MRD	CYP3A4 protects MRD from chemotherapy and targeted agents Liposomal chemotherapy (CPX351) is superior to “7 + 3”

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
