# Peer review of "MRD in Acute Leukemias: Lessons Learned from Acute Promyelocytic Leukemia"

_cancers, 2024, doi:10.3390/cancers16183208_

Round 1

Reviewer 1 Report

Comments and Suggestions for Authors

1. How Acute Promyelocytic Leukemia is different as compared with AML. Whether the flow markers were also different in both of these disorders.

2. What is the rate of OS, PFS, and DFS  of patients diagnosed with APL? It varies from population to population.

3. What is the relevance of blast percentage in APL?

4. Do patients with APL show any different clinical manifestations as compared with the normal cases?

5. Is there any molecular marker available to diagnose APL? NPM1, CEBPA, TET1, and FLT3 can also act as prognostic biomarkers in this disease.

Comments on the Quality of English Language

It seems fine.

Author Response

We appreciate the time and effort you have taken to review our manuscript. We have carefully considered your feedback and have made the following revisions to address each of your points:

  1. How Acute Promyelocytic Leukemia is different as compared with AML. Whether the flow markers were also different in both of these disorders.

We agree with you on the importance of diagnostic differences of APL and other non-APL AMLs. We have added a brief paragraph describing the unique immunophenotype of APL: “. Immunophenotypically, positivity of CD2, CD13, CD33, CD38, CD64, CD123, and MPO, and lack of CD34 and HLA-DR expression are highly suggestive for APL. Further details about flow cytometric diagnostic and differentials of APL have been discussed by Fang et al. [33].”

  1. What is the rate of OS, PFS, and DFS  of patients diagnosed with APL? It varies from population to population.

We agree that this is a very important point in epidemiology and outcomes of APL. In fact, the current manuscript is part of a Special Issue on APL. In this Special Issue, Sabine Kayser (DKFZ) and Shannon Conneely (Baylor College of Medicine) contributed a manuscript discussing “Management of APL at extremes of age”. In addition to this,  Amer Zeidan, Yale Medical School will contribute a manuscript addressing in depth current OS, PFS, DFS  and the important differences in these measurements between reports in clinical trials and SEER databases, between clinical care in academic centers and clinical care in community clinics. We made a point referencing these manuscripts in our current text and added the following: “APL accounts for 5-8% of AML patients with distinct biological, molecular, and clinical characteristics [29]. The epidemiology of APL has been thoroughly discussed in various epidemiological studies and reviews [30,31].”. 

  1. What is the relevance of blast percentage in APL?

Although the presenting WBC count has traditionally been a factor in APL risk stratification, this aspect is slowly losing prognostic value in the ATRA/ATO era. We made a point to mention this in the text: “While the presenting WBC count is part of the risk stratification of APL, the percentage of blasts are not formally part of any prognostic scoring system.”

  1. Do patients with APL show any different clinical manifestations as compared with the normal cases?

We appreciate the reviewers comment and we placed a reference in the text towards another manuscript from the Special Issue. This manuscript, authored by Bryan Hambley, University of Cincinnati focuses specifically on  “The Coagulopathy of Acute Promyelocytic Leukemia: An Updated Review of Pathophysiology, Risk Stratification, and Clinical Management”  and it discusses in details particularities of APL versus other AML cases: “Clinically, APL presents several particularities such as disseminated intravascular coag-ulation and other coagulopathies. These have been thoroughly discussed by Hermsen and Hambley [32].”

  1. Is there any molecular marker available to diagnose APL? NPM1, CEBPA, TET1, and FLT3 can also act as prognostic biomarkers in this disease.

Of course, presence of PML-RARA or t(15;17) is necessary to diagnose APL. There is currently lack of consensus that other genetic markers have any impact on management or prognosis of APL. We and others have shown that presence of FLT3ITD, particularly at high VAF may track with a relative worse prognosis (we have highlighted this controversy in the text). Since this is such an important topic that needs to be addressed in-depth, Maria Teresa Voso (Universita’ di Roma) will contribute a manuscript focused on the genetics of APL in this Special Issue.

Reviewer 2 Report

Comments and Suggestions for Authors

This is an interesting Review discussing the relevance of MRD in the clinical management of leukemia, and focusing on APL as a paradigm of targeted therapy. The work is well organized, clear and follows a rational flow during the chapters.

I propose some suggestions:

- Add figures or/and schemas to visually recap 1) methods to measure MRD 2) APL treatment

- when reporting “day 15 BM flow cytometry in ALL”, I suggest to add an important reference by Basso G et al, Risk of relapse of childhood acute lymphoblastic leukemia is predicted by flow cytometric measurement of residual disease on day 15 bone marrow. J ClinOncol. 2009 Nov 1; PMID: 19805690.

- Table 1 is not clear. The message is that some mutations are common between AML, MDS and CHIP, therefore I suggest to report a mutation for each row, and write if it is specific of a condition (for example only AML), or if it is present also in MDS or CHIP, to increase the clarity of the message conveyed.

- In the paragraph stating that in AML there is a lack of prospective MRD-directed clinical trials, it would be important to deep into NPM1mut, FLT3-ITD (treated in part when reporting giltertinib), and CBF, in particular AML1-ETO, for which there is a number of evidences sustaining the important contribution of MRD (Xu D et al, Risk-directed therapy based on genetics and MRD improves the outcomes of AML1-ETO-positive AML patients, a multi-center prospective cohort study, Blood Cancer Journal volume 13, Article number: 168 (2023); Pigazzi M, et al,. Minimal residual disease monitored after induction therapy by RQ-PCR can contribute to tailor treatment of patients with t(8;21) RUNX1-RUNX1T1 rearrangement. Haematologica. 2015;100:e99–101.; Zhu HH, et al. MRD-directed risk stratification treatment may improve outcomes of t(8;21) AML in the first complete remission: results from the AML05 multicenter trial. Blood. 2013;121:4056–62.; Hu GH, et al. Allogeneic hematopoietic stem cell transplantation can improve the prognosis of high-risk pediatric t(8;21) acute myeloid leukemia in first remission based on MRD-guided treatment. BMC Cancer. 2020;20:553.). Of note, MRD is a response to treatment criteria in the new AIEOP-BFM 2020 trial (“AML is high risk when minimal residual disease (MRD) is ≥1% after induction course 1 or ≥0.1% at induction 2”, Cacace F et al, High-Risk Acute Myeloid Leukemia: A Pediatric Prospective, Biomedicines. 2022 Jun; 10(6): 1405.)

- In the Table 2, in the following row:

Single-agent ATRA remission without cure

Targeting driver mutations alone is not enough to eliminate MRD

FLT3-, IDH-, menin-inhibitors cannot eliminate MRD as single agents

I do not agree that targeting the driver mutation alone is not enough to eliminate MRD, since ATRA overcomes the differentiation block induced by the PML-RARa fusion protein, but do not “target” driver mutation, as conversely ATO does (by inducing a proteasomal-mediated degradation of the oncoproptein). Consider to re-phrase this row.

- Additionally, in the Table 2, CYP26 is mentioned, but treated later in the text. I suggest to consider shifting the Table after the BME chapters.

- In the chapter 4.2: Considering the high relevance of the new immune-based therapies, I suggest to insert a paragraph regarding the BME role in mediating immunotherapy escape, that allow AML cells protection resulting in MRD persistence of leukemia re-emergence.

- Accordingly, in the chapter 4.3 immune reconstitution/ immune-based strategies are only barely treated, and should be examined more in depth.

- In the conclusion chapter: 1) I think that it would be more appropriate to write ATO+ATRA, instead of ATRA only, when discussing the targeting of driver mutations in APL. 2) Moreover, I do not agree with the sentence “We also know that single-agent ATRA is not effective in eliminating MRD in APL and targeting the microenvironment is essential to achieving a cure.”, as in APL ATO+ATRA led to an extremely high survival rate, so this is not the case in which “targeting the microenvironment is essential to achieving a cure”.

- The conclusion, in general, only considered APL, but should be enlarged including discussion on MRD, in accordance to the title.

Author Response

We appreciate your thorough review of our manuscript. Your insightful comments have been invaluable in improving the quality of our work. Below, we have addressed each of your concerns in detail.

  1. Add figures or/and schemas to visually recap 1) methods to measure MRD 2) APL treatment.

We created Figure 1 visualizing the MRD tools used in APL and AML in general and how they parallel changes in clinical management.

  1. When reporting “day 15 BM flow cytometry in ALL”, I suggest to add an important reference by Basso G et al, Risk of relapse of childhood acute lymphoblastic leukemia is predicted by flow cytometric measurement of residual disease on day 15 bone marrow. J ClinOncol. 2009 Nov 1; PMID: 19805690.

Thank you! We have added this important reference to the text.

  1. Table 1 is not clear. The message is that some mutations are common between AML, MDS and CHIP, therefore I suggest to report a mutation for each row, and write if it is specific of a condition (for example only AML), or if it is present also in MDS or CHIP, to increase the clarity of the message conveyed.

Thank you! We have updated Table 1.

  1. In the paragraph stating that in AML there is a lack of prospective MRD-directed clinical trials, it would be important to deep into NPM1mut, FLT3-ITD (treated in part when reporting gilteritinib), and CBF, in particular AML1-ETO, for which there is a number of evidences sustaining the important contribution of MRD (Xu D et al, Risk-directed therapy based on genetics and MRD improves the outcomes of AML1-ETO-positive AML patients, a multi-center prospective cohort study, Blood Cancer Journal volume 13, Article number: 168 (2023); Pigazzi M, et al,. Minimal residual disease monitored after induction therapy by RQ-PCR can contribute to tailor treatment of patients with t(8;21) RUNX1-RUNX1T1 rearrangement. Haematologica. 2015;100:e99–101.; Zhu HH, et al. MRD-directed risk stratification treatment may improve outcomes of t(8;21) AML in the first complete remission: results from the AML05 multicenter trial. Blood. 2013;121:4056–62.; Hu GH, et al. Allogeneic hematopoietic stem cell transplantation can improve the prognosis of high-risk pediatric t(8;21) acute myeloid leukemia in first remission based on MRD-guided treatment. BMC Cancer. 2020;20:553.). Of note, MRD is a response to treatment criteria in the new AIEOP-BFM 2020 trial (“AML is high risk when minimal residual disease (MRD) is ≥1% after induction course 1 or ≥0.1% at induction 2”, Cacace F et al, High-Risk Acute Myeloid Leukemia: A Pediatric Prospective, Biomedicines. 2022 Jun; 10(6): 1405.)

Thank you! We modified the paragraph discussing these aspects and cited the suggested papers: “Recent data clearly highlight the need and utility of MRD assessment in AML. Risk-adapted therapy, which takes into account the presence or absence of additional mutations in core binding factor (CBF) AML patients with translocations (8;11) and in-corporates MRD testing, has been shown to significantly improve outcomes in this sub-group [21,22]. The detection of RUNX1 in MRD testing was shown to correlate with negative outcomes in several studies [22,23]. More so, retrospective analysis of registry databases or of prospective randomized clinical trials clearly show that detection of somatic mutations via NGS at the time of transplant correlates with an increased risk of relapse [24,25]. This data suggests that perhaps more intensive conditioning regimens may erase the higher relapse rates in patients with MRD-positivity [24,25].

Furthermore, MRD positivity during first remission can help determine if a patient should be considered for allogeneic transplantation or guide the selection of post-transplant maintenance therapies [26]. For instance, data extrapolated from the phase 3 ADMIRAL trial (NCT02421939) comparing gilteritinib versus chemotherapy for re-lapsed AML patients showed significant superiority of gilteritinib maintenance after al-logeneic transplantation [27]. Subsequent, randomized clinical studies, however, reported that post-transplant gilteritinib is beneficial in MRD-positivity only, the progression-free and overall survival advantage being lost in MRD-negative patients [28]. These divergent observations highlight the fact that in the absence of randomized, MRD-directed clinical trials, the value of clinical observations should be interpreted with a healthy dose of academic skepticism.

In clinical practice, MRD detection in AML could lead to either relapse (70%) or cure (25-30%), however, it is prudent to be aware that the absolute quantifiable level of disease is not the sole determinant of patient outcomes. The biology of the disease, and other clinical factors modify the risk associated with MRD-test results. Despite the growing evidence on the role of MRD assessment in AML treatment, there remains a significant gap in our understanding of the biology of MRD. Thus, MRD-directed randomized clinical trial are necessary to bridge this gap.”

  1. In the Table 2, in the following row:

Single-agent ATRA remission without cure

Targeting driver mutations alone is not enough to eliminate MRD

FLT3-, IDH-, menin-inhibitors cannot eliminate MRD as single agents

I do not agree that targeting the driver mutation alone is not enough to eliminate MRD, since ATRA overcomes the differentiation block induced by the PML-RARa fusion protein, but do not “target” driver mutation, as conversely ATO does (by inducing a proteasomal-mediated degradation of the oncoproptein). Consider to re-phrase this row.

We have changed the word “targeting” to “inhibiting”. We believe that ATRA does inhibit the dominant negative effects on PML-RARA on differentiation block. Single agent ATRA obviously cannot completely eliminate MRD. On the other hand ATO does target the PML-RARA and perhaps has additional functions. That being said, single agent ATO is also not sufficient to completely eliminate MRD, though it is more efficient than ATRA. We have made the correction to the table and highlighted the controversy in the text.

  1. Additionally, in the Table 2, CYP26 is mentioned, but treated later in the text. I suggest to consider shifting the Table after the BME chapters.

 Done.

  1. In the chapter 4.2: Considering the high relevance of the new immune-based therapies, I suggest to insert a paragraph regarding the BME role in mediating immunotherapy escape, that allow AML cells protection resulting in MRD persistence of leukemia re-emergence. 

We think that immune therapies could bring a fresh perspective to treating AML. To emphasize the importance of immune escape resistance mechanisms, we’ve added the following paragraph: “Several mechanisms have been described through which the BME modulates both the innate and adaptive immune systems. In AML, the BM niche appears to block immune cell homing and enhances the migration of LSCs by altering the CXCL12-CXCR4 axis [97]. Additionally, MSCs have been shown to metabolically reprogram immune cells, inhibiting their activation, proliferation, and cytotoxic functions [98]. Spatial remodeling of the BM niches further hampers immune cell activity, creating a hypoxic, inflammatory, yet tumor-permissive environment [99, 100].”

  1. Accordingly, in the chapter 4.3 immune reconstitution/ immune-based strategies are only barely treated, and should be examined more in depth.

We appreciate you pointing this out. We added the following paragraph that analyzes currently tested drugs on immune reconstitution: “Similarly, robust immune reconstitution of the BM microenvironment plays an important role in the persistence of MRD in ALL and thus has a profound impact on the long-term relapse-free survival of these patients [105]. While immunotherapies have achieved remarkable clinical success in ALL, only a few antibody therapies, CAR T-cells, NK cells, or tumor vaccines have demonstrated activity against AML blasts. Several re-views have summarized the current clinical knowledge of these therapies and highlighted that at this time, the most promising targets in AML appear to be CD33 and CD123 [106-109].”

  1. In the conclusion chapter I think that it would be more appropriate to write ATO+ATRA, instead of ATRA only, when discussing the targeting of driver mutations in APL.

We agree on the importance of highlighting the crucial role of ATO, thus we modified the sentence “Yet, similar to ATRA in APL, we now see that targeting driver mutations in other AMLs differentiates malignant blasts and leads to differentiation syndrome.” to “Yet, similar to ATRA+ATO in APL, we now see that targeting driver mutations in other AMLs differentiates malignant blasts and leads to differentiation syndrome.”.

  1. In the conclusion chapter I do not agree with the sentence “We also know that single-agent ATRA is not effective in eliminating MRD in APL and targeting the microenvironment is essential to achieving a cure.”, as in APL ATO+ATRA led to an extremely high survival rate, so this is not the case in which “targeting the microenvironment is essential to achieving a cure”.

We agree that this statement is speculative at this point. Thus, we have modified it to: “We also know that single agent ATRA is not effective in eliminating MRD in APL likely due to the BME – dependent mechanisms. To what extent the success of ATRA+ATO combination in eliminating MRD directly addresses this mechanism of resistance remains to be explored.”

  1. The conclusion, in general, only considered APL, but should be enlarged including discussion on MRD, in accordance to the title.

We added some future perspectives on MRD to the conclusion: “Building on our current knowledge, future AML research should prioritize advancing MRD detection through novel assessment tools, identifying new mutations for MRD monitoring, and developing MRD prognostic scoring systems. Thus, developing novel drug combinations that target the bone marrow microenvironment hold the potential to bypass stroma-mediated protection of LSCs and stimulate the immune system to eliminate MRD and improve long-term outcomes in patients with AML.”

Round 2

Reviewer 2 Report

Comments and Suggestions for Authors

Line 157: t(8;21) instead of t(8;11), and incorporate instead of in-corporates.

 Line 159: the detection of RUNX1::RUNX1T1, not RUNX1.

Line 441: correct re-views.  

I really do not understand the answer to question 10, when the authors said: “To what extent the success of ATRA+ATO combination in eliminating MRD directly addresses this mechanism of resistance remains to be explored.” APL are the only AML with a target therapy that degrades the oncogene, thus this leukemia is cured with ATRA+ATO, and the mechanism is well described (PML-RARA degradation via proteasoma). I don’t understand why authors said that the mechanism remains to be explored.

The chapter 4.2.4 has been added in response to my comment, but not really discussed. I suggest to enlarge it and treat the field of BME contribution to immune escape of LSCs more in depth.

Author Response

Line 157: t(8;21) instead of t(8;11), and incorporate instead of in-corporates. Line 159: the detection of RUNX1::RUNX1T1, not RUNX1. Line 441: correct re-views.  

Thank you for these observations, we corrected them. 

The chapter 4.2.4 has been added in response to my comment, but not really discussed. I suggest to enlarge it and treat the field of BME contribution to immune escape of LSCs more in depth.

Thank you for your feedback. We expanded the paragraph and provided additional reviews and resource though an in depth analysis of the immune escape is beyond the scope of the manuscript in keeping in accordance with weight on another mechanisms described and the overall word constrains. The new subchapter is the following: “It was early on recognized that immune escape is a major mechanism responsible for persistence of MRD. In fact, most recent bone marrow transplant trends in which allogeneic effect is necessary for elimination of MRD are exemplified by the fact that autologous transplant has no role in consolidation of AML. Thus, the most effective and widely used therapy to eliminate MRD in AML remains the immune therapy afforded by allogeneic transplantation. Although various mechanisms have been described in the literature, the exact ones responsible for immune escape in MRD are still not fully understood [97]. That being said, data from studies of the cellular and molecular composition of the stem cell niche point towards some potential mechanisms by which the BME may create immune sanctuaries that can protect MRD. For instance, regulatory T-cell and myeloid derived dendritic cells are important constituents of the stem cell niche and thus, could protect malignant cells from an otherwise effective immune attack [98]. Also, in AML, the BM niche appears to block immune cell homing and enhances the migration of LSCs by altering the CXCL12-CXCR4 axis, metabolically reprogram immune cells, inhibiting their activation, proliferation, and cytotoxic functions [99, 100]. Spatial remodeling of the BM niches further hampers immune cell activity, creating a hypoxic, inflammatory, yet tumor-permissive environment [101, 102]. 

I really do not understand the answer to question 10, when the authors said: “To what extent the success of ATRA+ATO combination in eliminating MRD directly addresses this mechanism of resistance remains to be explored.” APL are the only AML with a target therapy that degrades the oncogene, thus this leukemia is cured with ATRA+ATO, and the mechanism is well described (PML-RARA degradation via proteasoma). I don’t understand why authors said that the mechanism remains to be explored.

Regarding the MRD elimination comment, we have tried to further clarify in the manuscript that while ATRA+AT0 is synergistic on the APL cells by degrading the PML-RARA, it was not formally shown that this exact mechanism is also responsible for the elimination of MRD. This is in keeping with the lack of formal data showing the molecular mechanism by which ATO eliminates MRD in APL. We added the following paragraph to explain clarify: “While it is likely that ATO degrades the PML-RARA in APL MRD, other mechanisms of actions have also been described for ATO, including induction of ROS, promotion of autophagy, inhibition of Hedgehog signaling [117-119]. To what extent these mechanisms play a role in the success of ATRA+ATO combination in eliminating MRD remains to be explored.”